# DE NOVO DRUG DESIGN WITH JOINT TRANSFORMERS

## ABSTRACT

*De novo* drug design requires simultaneously generating novel molecules outside of training data and predicting their target properties, making it a hard task for generative models. To address this, we propose JOINT TRANSFORMER that combines a Transformer decoder, Transformer encoder, and a predictor in a joint generative model with shared weights. We show that training the model with a penalized log-likelihood objective results in state-of-the-art performance in molecule generation, while decreasing the prediction error on newly sampled molecules, as compared to a fine-tuned decoder-only Transformer, by $42\%$. Finally, we propose a probabilistic black-box optimization algorithm that employs JOINT TRANSFORMER to generate novel molecules with improved target properties, as compared to the training data, outperforming other SMILES-based optimization methods in *de novo* drug design.

## 1 INTRODUCTION

*De novo* drug design is an approach to generate novel structures with desired properties from scratch. It opens the door to new classes of drugs, promising to overcome limitations of existing treatments (Schneider & Clark, 2019). While numerous breakthroughs in generative modeling and natural language processing (Vaswani et al., 2017; Radford et al., 2018) advanced the field of drug discovery, *de novo* design remains a notoriously challenging task (Wu et al., 2021; Grisoni, 2023).

*De novo* design requires to simultaneously (i) generate novel compounds, (ii) accurately predict their target properties and (iii) optimize the generation of compounds towards the desired properties (Brown et al., 2019). However, as the desired properties are rarely observed in the training data, there is an inherent trade-off between generation, prediction, and optimization. The more optimized towards properties from outside the training distribution the generation is, the less reliable the generation of compounds and prediction of their properties become.

Previous work on generative models for *de novo* drug design focused on each of the required components separately. Decoder-only Transformers (Radford et al., 2018) successfully generate novel and chemically plausible molecules (Bagal et al., 2022), but they have no information about the target properties. These can be fine-tuned or coupled with RL approaches, however, without yielding satisfactory results in practical regimes (Neil et al., 2018). Encoder-only Transformers excel at molecular property prediction tasks (Ross et al., 2022; Zhou et al., 2023b), but they lack molecule generation capabilities. Optimization of molecules is often treated as a black-box optimization (BBO) problem (Terayama et al., 2021) and solved over a continuous latent space of a Latent Variable Model like a Variational Autoencoder (Kingma & Welling, 2013; Rezende et al., 2014), using an external optimization routine, e.g., Bayesian Optimization (Gómez-Bombarelli et al., 2018; Tripp et al., 2020). However, posing the problem as BBO and employing an external optimization routine tend to guide a generative model far from the true data distribution of chemically plausible molecules, in regions where the generation and prediction becomes unreliable. The lack of a coherent framework that would address all the above challenges at the same time motivates the need for a joint approach.

In this paper, we propose JOINT TRANSFORMER, a joint generative model that simultaneously generates novel examples and accurately predicts their target properties. We achieve this by combining a Transformer decoder (driving the generative performance) with a Transformer encoder and a predictor (both encouraging predictive performance). We propose to train the joint generative model with a penalized log-likelihood objective, which allows simultaneous training of the decoder, encoder, and predictor, enabling joint training and sharing all the weights. Equipped with JOINT TRANSFORMER, we pose *de novo* drug design as a probabilistic version of the BBO problem, where

we aim to optimize a given objective function, like in BBO, but only in the regions of the input space, where the generative model assigns high likelihood to samples and the generative and predictive capabilities of the model remain reliable. Finally, we propose a sampling algorithm that utilizes the strong generative and predictive performance of the model to generate optimized molecules.

The contribution of the paper is threefold: **(i)** We propose JOINT TRANSFORMER, a joint generative model that simultaneously generates novel examples and accurately predicts their target properties (Section 3.1). **(ii)** We formulate a generic sampling algorithm with theoretical guarantees (Section 3.1.3) to guide the generation of novel compounds with JOINT TRANSFORMER. **(iii)** We show that JOINT TRANSFORMER outperforms standard approaches for fine-tuning generative models (Section 4.1), successfully finds molecules with a high biological affinity toward a given target from a given library (Section 4.2) and outperforms state-of-the-art SMILES-based optimization methods in the *de novo* drug design task (Section 4.3).

## 2 METHODOLOGY

### 2.1 PROBLEM STATEMENT

Let us consider an optimization problem where, given an *objective function* $f : \mathcal{X} \to \mathbb{R}$, the goal is to find examples $\mathbf{x}^* \in \mathcal{X}$ that maximize the objective function $f$, namely:

$$\mathbf{x}^* = \arg\max_{\mathbf{x} \in \mathcal{X}} f(\mathbf{x}). \tag{1}$$

In the *black-box* optimization (BBO) setting (Alarie et al., 2021; Audet & Hare, 2017; Terayama et al., 2021), we assume no analytical form of the objective $f$. In particular, $f$ may not be differentiable.

Direct optimization of the objective $f$, over the whole input space $\mathcal{X}$, may lead to examples $\mathbf{x} \in \mathcal{X}$ that are not expected to be observed and are not meaningful (Brookes et al., 2019; Renz et al., 2019). To account for this, we define an example $\mathbf{x} \in \mathcal{X}$ as *semantically meaningful*, if $\mathbf{x}$ could have been generated by the true data generating distribution $p(\mathbf{x})$.

In order to constrain the optimization problem to *meaningful* examples, we propose to treat BBO in a probabilistic manner. Consider a target $y \in \mathbb{R}$ defined by the objective function $f(\mathbf{x}) = y$, for an example $\mathbf{x} \in \mathcal{X}$. We define *probabilistic* BBO (PBBO) as the problem of sampling examples $\mathbf{x}^* \in \mathcal{X}$ maximizing the objective function $f$ that could have been generated by the true underlying data distribution, i.e., they are semantically meaningful:

$$\mathbf{x}^* \sim p(\mathbf{x} \mid y_{\max}), \text{ where } y_{\max} = \max_{\mathbf{x} \sim p(\mathbf{x})} f(\mathbf{x}). \tag{2}$$

We claim that the problem of *de novo* drug design is an instance of a PBBO problem. In *de novo* drug design the objective function $f$ is the outcome of a laboratory experiment measuring a molecular property of interest, and the input space $\mathcal{X}$ is the space of all possible molecules. As a large fraction of the input space $\mathcal{X}$ corresponds to molecules that are non-meaningful for drug design purposes, e.g., molecules that are represented by valid SMILES strings but cannot be synthesized, we postulate that the true underlying data distribution of known molecules $p(\mathbf{x})$ is the anchor for optimizing only over meaningful drug candidates. One potential way of estimating $p(\mathbf{x})$ from data is generative modeling.

### 2.2 GENERATIVE MODELING

The goal of generative modeling is to learn a probability distribution (either a joint distribution, a marginal distribution, or a conditional distribution) given training data (Tomczak, 2021). Since the problem stated in Eq. 2 requires learning a conditional probability distribution, generative modeling seems to be a perfect fit. However, typically, in *de novo* drug design, we must face targets $y$ that are continuous, thus, learning conditional distributions becomes more complicated (i.e., we must learn an infinite mixture of distributions instead of a finite mixture). Moreover, in practice, we can access only a small subset of examples with observed values of a target $y$.

**Unconditional generative models** The problem stated in Eq. 2 can be approached by considering a subset of examples $\mathbf{x} \in \mathcal{X}$ such that $f(\mathbf{x}) \geq y_c$, where $y_c \in \mathbb{R}$ is a predefined threshold. First,

a generative model is pre-trained using all available examples and then fine-tuned using only the subset of examples such that $f(\mathbf{x}) \geq y_c$ (Grisoni, 2023). This procedure results in an unconditional generative model $p_\theta(\mathbf{x})$ approximating the conditional distribution $p(\mathbf{x} \mid y \geq y_c)$. As the generative model $p_\theta(\mathbf{x})$ is trained in an unsupervised manner, it gives no indication whether a newly sampled example $\mathbf{x} \sim p_\theta(\mathbf{x})$ enjoys a high value of the objective $f(\mathbf{x})$. Moreover, the event $f(\mathbf{x}) \geq y_c$ tends to be rare, leaving little to no examples $\mathbf{x}$ for finetuning the generative model $p_\theta(\mathbf{x})$ (Brookes et al., 2019). Finally, a separate model $p_\theta(\mathbf{x})$ is needed for every target threshold $y_c \in \mathbb{R}$.

**Latent Space Optimization**    An alternative approach relies on using a latent variable model (LVM), typically a VAE (Kingma & Welling, 2013; Rezende et al., 2014), to learn a joint distribution $p(\mathbf{x}, \mathbf{z}, y)$, and then utilizing a latent space $\mathcal{Z}$ to carry out property optimization (Gómez-Bombarelli et al., 2018). This method is referred to as Latent Space Optimization (LSO). As for sampling, LSO employs an external optimization routine (e.g., Bayesian Optimization) to choose a latent vector $\mathbf{z} \in \mathcal{Z}$ that upon decoding with the LVM's decoder $p_\theta(\mathbf{x} \mid \mathbf{z})$ will result in an example $\mathbf{x} \in \mathcal{X}$ enjoying a high value of the objective $f(\mathbf{x})$. However, sampling using an external optimization routine often results in choosing a latent point $\mathbf{z}$ out of the training distribution of the latent variable model, leading to either a decoded example $\mathbf{x}$ that is not semantically meaningful or the decoder $p(\mathbf{x} \mid \mathbf{z})$ ignoring the latent vector $\mathbf{z}$. In a similar manner, diffusion-model-based methods (Hoogeboom et al., 2022), which learn $p(\mathbf{x} \mid y)$ directly, face the practical problem of the decoder ignoring the target $y$.

**Transformer-based models**    Transformers (Vaswani et al., 2017) are state-of-the-art models across both generative modeling and representation learning tasks, including molecule generation (Bagal et al., 2022) and molecular representation learning (Zhou et al., 2023a). Different tasks require using different Transformer-based models, with Transformer decoders (e.g., GPT (Brown et al., 2020)) being typically used for generation and Transformer encoders (e.g., BERT (Devlin et al., 2018)) for representation learning.

The training procedure of a Transformer decoder, which is an autoregressive model (ARM) that samples one token at a time, amounts to minimizing the negative log-likelihood:

$$\ell(\theta) = -\mathbb{E}_{\mathbf{x} \sim p(\mathbf{x})} \left[ \ln p_\theta(\mathbf{x}) \right]. \tag{3}$$

The training procedure of a Transformer encoder starts with drawing a random vector $\mathbf{m} = (m_1, \ldots, m_D) \sim q(\mathbf{m})$, with $m_d$ indicating whether token $x_d$ is masked out. In practice, each token is masked out with a fixed probability (Devlin et al., 2018). The training procedure amounts to minimizing the negative pseudo-log-likelihood (Besag, 1975) over tokens $\mathbf{x}_{-d}$:

$$\ell(\theta) = -\mathbb{E}_{\mathbf{x} \sim p(\mathbf{x})} \left\{ \mathbb{E}_{\mathbf{m} \sim q(\mathbf{m})} \left[ \sum_{d=1}^{D} \ln p_\theta(x_d \mid \mathbf{m} \odot \mathbf{x}_{-d}) \right] \right\}. \tag{4}$$

Note that the Transformer decoder is an ARM with a self-attention mechanism that incorporates *causal masking* of the input sequence, i.e., for each token, all future tokens are masked out from computations. On the other hand, a Transformer encoder incorporates *bidirectional masking*, which includes all tokens into the computations, i.e., treats all tokens in the sequence as the context. The difference between the distribution $\prod_{d=1}^{D} p_\theta(x_d \mid \mathbf{m} \odot \mathbf{x}_{-d})$ and $p_\theta(\mathbf{x})$ lies in the choice of masking used together with the self-attention layer, bidirectional masking in $\prod_{d=1}^{D} p_\theta(x_d \mid \mathbf{m} \odot \mathbf{x}_{-d})$, and causal masking in $p_\theta(\mathbf{x})$.

To practically enable weight sharing between a Transformer decoder and a Transformer encoder, the training steps in Eq. 3 and in Eq. 4 are combined, similarly to (Dong et al., 2019), resulting in the penalized negative log-likelihood function:

$$\ell(\theta) = -\mathbb{E}_{\mathbf{x} \sim p(\mathbf{x})} \left\{ \ln p_\theta(\mathbf{x}) + \mathbb{E}_{\mathbf{m} \sim q(\mathbf{m})} \left[ \sum_{d=1}^{D} \ln p_\theta(x_d \mid \mathbf{m} \odot \mathbf{x}_{-d}) \right] \right\}. \tag{5}$$

Typically, Transformers in the context of molecular modeling are used to learn the distribution $p_\theta(\mathbf{x})$ for molecules. Then, they are utilized for an unconditional proxy to $p(\mathbf{x} \mid y)$. As a result, they suffer similar issues as unconditional generative models.

## 3  OUR APPROACH

To address the problem of learning and sampling from the conditional distribution $p(\mathbf{x} \mid y)$ in Eq. 2, we propose a joint generative model of examples and corresponding targets. The advantage of such an approach is twofold. First, joint modeling encourages sharing the weights used for generation and prediction, making robust prediction of target values on newly generated examples feasible. Second, the robust predictions give a good indication of whether the newly generated examples have high values of the target. Indeed, the joint generative model allows sampling examples $\mathbf{x}$ that fulfill Eq. 2 and satisfy the desired condition $y \geq y_c$, for every $y_c \in \mathbb{R}$.

### 3.1  JOINT TRANSFORMER

The proposed joint generative model, JOINT TRANSFORMER, $p_{\theta,\phi}(\mathbf{x}, y)$ combines three models: a Transformer decoder $p_\theta(\mathbf{x})$, a Transformer encoder $\prod_{d=1}^{D} p_\theta(x_d \mid \mathbf{m} \odot \mathbf{x}_{-d})$, and a predictor $p_{\theta,\phi}(y \mid \mathbf{x})$. The weights $\theta$ are shared between the encoder, decoder, and predictor parts. Additionally, the predictor (used either for regression or classification) is stacked on the top of the encoder and is parametrized with weights $\phi$. The difference between the decoder and the encoder lies only in the choice of masking used within attention layers, namely, the decoder uses causal masking while the encoder applies bidirectional masking.

The rationale behind our model is the following. First, we share weights $\theta$ to entangle the generation and prediction tasks and make robust predictions of target values on newly generated examples feasible. At the same time, sharing weights has the practical advantage of a more computationally efficient model. Second, alongside the Transformer decoder, we incorporate the Transformer encoder to let the predictor learn better representations and process the input in a non-sequential manne, as a lack of bidirectional context may be harmful to predictive performance (Devlin et al., 2018).

#### 3.1.1  TRAINING

In order to learn a single model that combines a Transformer encoder, a Transformer decoder, and a predictor, we propose to minimize a penalized negative log-likelihood of the joint model given by:

$$\ell(\theta, \phi) = -\mathbb{E}_{(\mathbf{x},y)\sim p(\mathbf{x},y)} \Bigg\{ \ln p_\theta(\mathbf{x}) + \ln p_{\theta,\phi}(y \mid \mathbf{x}) + $$

$$\mathbb{E}_{\mathbf{m}\sim q(\mathbf{m})} \left[ \sum_{d=1}^{D} \ln p_\theta(x_d \mid \mathbf{m} \odot \mathbf{x}_{-d}) \right] \Bigg\}, \tag{6}$$

where $q(\mathbf{m})$ is an arbitrary masking distribution. Using the penalized negative log-likelihood objective (Eq. 6) encourages the model to simultaneously operate in two separate modes: input generation and property prediction. First, updating the decoder and learning to process the input in an autoregressive manner drives the generative performance of the model. Second, updating the encoder and learning to process the input in a bidirectional manner drives learning a meaningful representation and the predictive performance, since the predictor shares weights $\theta$ with the encoder.

Training a joint generative model was previously shown to result in a good generator together with a poor predictor (Lasserre et al., 2006; Nalisnick et al., 2019). Moreover, storing the gradients for all the summands of the objective (Eq. 6) is a significant overhead in memory requirements as compared to decoder and encoder-only Transformers. To overcome both issues, we propose a practical training procedure for JOINT TRANSFORMER (Alg. 1) that randomly switches between the input generation and the property prediction (and encoder training) tasks with a hyperparameter $p_{\text{task}} \in [0, 1]$.

The JOINT TRANSFORMER can be trained in an unsupervised, semi-supervised or supervised setting. Depending whether a target $y \in \mathcal{Y}$ is sampled from the dataset $\mathcal{D}$ or is not available (Step 2, Alg. 1), one can include the prediction loss $\ln p_{\theta,\phi}(y \mid \mathbf{x})$ in the penalized log-likelihood objective $\ell$ (Step 6, Alg. 1), resulting in a supervised setting, or set the prediction loss to zero, resulting in an unsupervised setting. For the training data where only a small proportion of samples have accompanying target values, we split the training procedure of the JOINT TRANSFORMER into first training the model in an unsupervised manner (Alg. 3 in Appendix D.4), then fine-tuning it with supervised data (Alg. 1).

---

**Algorithm 1** Training of JOINT TRANSFORMER

---

**Input:** A dataset $\mathcal{D} = \{(\mathbf{x}_n, y_n)\}_{n=1}^N$. JOINT TRANSFORMER $p_{\theta,\phi}(\mathbf{x}, y)$ with parameters $\theta, \phi$
    containing a decoder $p_\theta(\mathbf{x})$, encoder $\prod_{d=1}^D p_\theta(x_d \mid \mathbf{m} \odot \mathbf{x}_{-d})$ and a predictor $p_{\theta,\phi}(y \mid \mathbf{x})$.
    Task probability $p_{\text{task}} \in [0, 1]$ and a masking distribution $q(\mathbf{m})$.

  1: **while** a stopping criterion is not met **do**
  2:     Uniformly sample $(\mathbf{x}, y)$ from the dataset $\mathcal{D}$
  3:     Sample an indicator $u \sim \text{BERNOULLI}(p_{task})$
  4:     **if** $u = 0$ **then**
  5:         Sample mask $\mathbf{m} \sim q(\mathbf{m})$
  6:         Calculate loss $\ell(\theta, \phi) = -\sum_{d=1}^D \ln p_\theta(x_d \mid \mathbf{m} \odot \mathbf{x}_{-d}) - \ln p_{\theta,\phi}(y \mid \mathbf{x})$
  7:     **else**
  8:         Set mask to the causal mask
  9:         Calculate loss $\ell(\theta, \phi) = -\ln p_\theta(\mathbf{x})$
10:     **end if**
11:     Update parameters $\theta, \phi$ using an optimizer w.r.t. loss $\ell$
12: **end while**

---

### 3.1.2 UNCONDITIONAL GENERATION

In the unconditional generation task, we sample from JOINT TRANSFORMER in a two-step manner that results in an unconditional sample $(\mathbf{x}, y) \sim p_{\theta,\phi}(\mathbf{x}, y)$. First, since the decoder part $p_\theta(\mathbf{x})$ does not depend on parameters $\phi$ and it properly defines an ARM, we sample $\mathbf{x} \sim p_\theta(\mathbf{x})$. Next, we sample a target $y$ from the predictive distribution $y \sim p_{\theta,\phi}(y \mid \mathbf{x})$. The key feature that allows for successful sampling from the joint model is the ability of the JOINT TRANSFORMER to simultaneously operate in two separate modes, namely generate novel examples and predict their target values, which is directly encouraged by training with the penalized log-likelihood objective in Eq. 6.

### 3.1.3 CONDITIONAL GENERATION

In the conditional generation task, given a condition $Y \subseteq \mathcal{Y}$, we sample from JOINT TRANSFORMER $p_{\theta,\phi}(\mathbf{x}, y)$ to obtain a conditional sample $(\mathbf{x}, y) \sim p_{\theta,\phi}(\mathbf{x}, y)$, such that $y \in Y$. JOINT TRANS-FORMER generates conditional samples by first sampling $(\mathbf{x}, y) \sim p_{\theta,\phi}(\mathbf{x}, y)$ in the above described unconditional way and then accepting the sample if $y \in Y$. In practical applications, due to a finite runtime, we sample a batch of $B$ tuples $(\mathbf{x}, y) \sim p_{\theta,\phi}(\mathbf{x}, y)$ and choose $(\mathbf{x}, y)$ with $y$ 'closest' to $Y$. Proposition 1 shows that, despite its conceptual simplicity, the described conditional generation procedure is equivalent to directly sampling from the conditional distribution $p(\mathbf{x} \mid y)$. Moreover, Proposition 2 shows conditions under which conditional generation enjoys a finite expected runtime.

**Proposition 1.** *Let $p(\mathbf{x}, y)$ be a joint probability distribution over $\mathcal{X} \times \mathcal{Y}$. Let $y_c \in \mathcal{Y}$ be such that $p(y_c) > 0$. Then*

$$p(\mathbf{x} \mid y_c) \propto \mathbb{1}_{\{y=y_c\}}(y)p(y \mid \mathbf{x})p(\mathbf{x}).$$

*Proof.* See Appendix B.1.        □

**Proposition 2.** *Let $p(y)$ be a probability distribution over $\mathcal{Y}$ with a corresponding cumulative distribution function $F$. Let target $y_c \in \mathcal{Y}$ be such that $p(y_c) > 0$ and let $p$ be the probability of sampling a target $y \sim p(y)$ such that $y > y_c$. The expected number of trials $N$ until obtaining a sample $y \sim p(y)$ such that $y > y_c$ is equal to $1/p$.*

*Proof.* See Appendix B.2.        □

Despite its simplicity, the conditional generation of JOINT TRANSFORMER has the advantage of the predictor $p_{\theta,\phi}(y \mid \mathbf{x})$, as it is defined in the input space $\mathcal{X}$, indicating whether the newly generated example enjoys the desired target value. This is in contrast to methods based on LSO and diffusion models, see Section 2.2.

### 3.2 Probabilistic Black Box Optimization

We define PBBO as the problem of sampling from the conditional distribution $\mathbf{x} \sim p(\mathbf{x} \mid y_c)$, where target $y_c \in \mathbb{R}$ is equal or close to the optimal value of the objective function $f$ (Section 2.1). However, in Proposition 1 we show that sampling $\mathbf{x} \sim p(\mathbf{x} \mid y_c)$ is equivalent to the conditional generation from a joint model like JOINT TRANSFORMER. Moreover, in Proposition 2 we show that conditional sampling is practically feasible, as long as target $y_c$ is within the support of JOINT TRANSFORMER. In practice, to fix a feasible threshold $y_c$, one can set a sampling budget of $B \in \mathbb{N}$ examples and then rank examples according to $p(y \mid \mathbf{x})$ choosing the best examples available. Algorithm 3.2 shows how to facilitate PBBO using JOINT TRANSFORMER.

---

**Algorithm 2** Probabilistic Black-Box Optimization with JOINT TRANSFORMER

---

**Input:** JOINT TRANSFORMER $p_{\theta,\phi}(\mathbf{x}, y)$ with parameters $\theta, \phi$.
    Threshold $y_c \in \mathcal{Y}$. Evaluation budget $I \in \mathbb{N}$. Sampling budget $B \in \mathbb{N}$.
**Output:** $\mathcal{D}_{\text{new}} = \{(\mathbf{x}_i, y_i)\}_{i=1}^{min(I,B)}$
 1: $b \leftarrow 0, i \leftarrow 1$
 2: **while** $b < B$ **and** $i \leq I$ **do**
 3:    Unconditionally sample $(\mathbf{x}_i, y_i) \sim p_{\theta,\phi}(\mathbf{x}, y)$
 4:    **if** $y_i \geq y_c$ **then**
 5:        $\mathcal{D}_{\text{new}} \leftarrow \mathcal{D}_{\text{new}} \cup \{(\mathbf{x}_i, y_i)\}$
 6:        $i \leftarrow i + 1$
 7:    **end if**
 8:    $b \leftarrow b + 1$
 9: **end while**

---

The proposed optimization procedure addresses the needs of *de novo* drug design in several ways. First, it is probabilistic and therefore tailored to avoid non-realistic molecules as sampled examples. Second, it gives a guarantee of sampling improved examples. Third, it deals with the issue of a low fraction of labeled training data (low fraction of known target values for the molecules) by incorporating unsupervised training phases.

## 4 Experiments

To validate the applicability of JOINT TRANSFORMER to molecule data, we first demonstrate that joint training allows JOINT TRANSFORMER to supersede standard fine-tuning approaches and that JOINT TRANSFORMER successfully samples novel examples with corresponding target values, performing joint unconditional generation (Section 4.1). Second, we show that JOINT TRANSFORMER outperforms standard machine-learning models equipped with features extracted from pre-trained chemical models in a real-world classification task (Section 4.2). Finally, we show that JOINT TRANSFORMER combined with the probabilistic Black Box Optimization algorithm (Alg. 3.2) outperforms alternative methods for generating optimized molecules, as illustrated by application to a *de novo* drug design task (Section 4.3).

We choose the architecture of GPT (Radford et al., 2018) for JOINT TRANSFORMER in all tasks. The same architecture was previously utilized in MolGPT by Bagal et al. (2022). However, the training of our model is different from Bagal et al. (2022), and follows Alg. 1. Implementation details for JOINT TRANSFORMER are outlined in Appendix D. Although our evaluation showed similarly high performance of JOINT TRANSFORMER and MolGPT in molecule generation (Appendix C.1), JOINT TRANSFORMER shows a clear advantage over MolGPT in the predictive and molecule optimization tasks, as outlined below. In all tasks we use a JOINT TRANSFORMER pre-trained, in an unsupervised manner, using molecules derived from the ChEMBL 24 dataset (Mendez et al., 2019), following Brown et al. (2019) for processing and splitting the data. In the ablation study and *de novo* design task we additionally fine-tune JOINT TRANSFORMER using penalized log-likelihood on randomly selected subsets of the training data ($N = 1000$) with multi-property objective functions (MPO), derived from the GuacaMol benchmark (Brown et al., 2019), as continuous targets in the range $[0, 1]$.

### 4.1 ABLATION STUDY (UNCONDITIONAL GENERATION & PREDICTION)

**Task** First, we compare the generative and predictive performance of the JOINT TRANSFORMER to its ablated versions, with the aim to determine the influence of the different components of the penalized negative log-likelihood objective (Eq. 6). First, we evaluate the generative performance using three metrics: validity, the fraction of the generated molecules that correspond to syntactically valid SMILES strings; KL Divergence, a measure of similarity between the distribution of the selected chemical properties of the generated and training molecules, as well as Fréchet ChemNet Distance (FCD; (Preuer et al., 2018)), a general measure of similarity of the generated molecules to the training set. Second, we evaluate the predictive performance using MAE, the mean absolute error calculated on the test set of the Guacamol dataset, and MAE$_{\text{SAMPLED}}$, calculated on novel molecules generated with a particular version of the model, with Zaleplon MPO being the continuous target.

**Models** We compare two ablated versions of the JOINT TRANSFORMER. JOINT TRANSFORMER without penalty and without $\ln p_\theta(\mathbf{x})$, JT W/O PENALTY W/O $\ln p_\theta(\mathbf{x})$, corresponds to a model pre-trained without the reconstruction penalty and fine-tuned using only the predictive component $\ln p_{\theta,\phi}(y \mid \mathbf{x})$. Next, JOINT TRANSFORMER without penalty, JT W/O PENALTY, corresponds to a model pre-trained without the reconstruction penalty and fine-tuned using the predictive and the generative components $\ln p_{\theta,\phi}(y \mid \mathbf{x}) + \ln p(\mathbf{x})$. Additionally, we compare JOINT TRANSFORMER fine-tuned with two different values of the task probability $p_{\text{task}} \in \{0.1, 0.5\}$, which trades off the generative and predictive performance of the JOINT TRANSFORMER. We fine-tune all models on randomly selected subsets of the training data ($N = 1000$).

**Results** Fine-tuning JT W/O PENALTY W/O $\ln p_\theta(\mathbf{x})$ results in good predictive performance, in terms of low MAE, but also results in forgetting the molecule generation task, as the validity of the generated molecules drops to zero (Table 1). Importantly, this is the standard way of fine-tuning language models (Raffel et al., 2020), and is aimed to maximize the predictive performance. An approach to prevent forgetting the molecule generation task is to fine-tune the model only without penalty, as it is done for JT W/O PENALTY (Brown et al., 2020). Although fine-tuning JT W/O PENALTY indeed obtains satisfactory validity, it comes at the price of worsening the predictive performance of the model. In contrast, JOINT TRANSFORMER trained with the full penalized log-likelihood achieves the best predictive performance without notably sacrificing the generative ability. Particularly, the JOINT TRANSFORMER fine-tuned with $p_{\text{task}} = 0.1$ outperforms the standard fine-tuned JT W/O PENALTY W/O $\ln p_\theta(\mathbf{x})$ in predictive performance by a large margin, showing the positive effect of joint training. In summary, JOINT TRANSFORMER outperforms standard fine-tuning alternatives and successfully performs unconditional generation.

Table 1: Ablation study of various training loss functions and different values of $p_{task}$. Mean and standard deviation across three data subsets. All models achieve KL Divergence equal to $0.99 \pm 0.00$.

| MODEL | VALIDITY ($\uparrow$) | FCD ($\uparrow$) | MAE($\downarrow$) | MAE$_{\text{SAMPLED}}$($\downarrow$) |
|---|---|---|---|---|
| JT W/O PENALTY W/O $\ln p_\theta(\mathbf{x})$ | $0.00 \pm 0.00$ | N/A | $0.016 \pm 0.003$ | N/A |
| JT W/O PENALTY | $0.97 \pm 0.01$ | $0.87 \pm 0.01$ | $0.021 \pm 0.001$ | $0.023 \pm 0.001$ |
| JT ($p_{\text{task}} = 0.5$) (OURS) | $0.97 \pm 0.00$ | $0.86 \pm 0.00$ | $0.019 \pm 0.002$ | $0.019 \pm 0.001$ |
| JT ($p_{\text{task}} = 0.1$) (OURS) | $0.96 \pm 0.00$ | $0.85 \pm 0.01$ | $0.012 \pm 0.001$ | $0.012 \pm 0.001$ |

### 4.2 TARGETED VIRTUAL SCREENING

**Task** The targeted screening task is a classification task with the goal of selecting molecules from a given library with a high biological affinity toward a given target. In this experiment, the target is human alanyl aminopeptidase (hsAPN), for which we obtained 590 training molecules from (Liu et al., 2007) and 48 test molecules from (Vassiliou et al., 2014; Weglarz-Tomczak et al., 2016). We turn available properties (either $IC_{50}$ or $K_i$) into binary labels (corresponding to active and inactive molecules). In training data, there are 68% of inactive molecules, while in test data there are 35% of inactive molecules. This setting mimics a real-life application in which there is a scarce amount of training data in the target domain. Also, it is likely that test molecules vary structurally from train molecules. We evaluate all models using precision, recall, F1, and accuracy.

**Models**   We compare the fine-tuned (using the training data from the targeted virtual screening task) JOINT TRANSFORMER to various combinations of predictors and Chemical Pre-trained Models (CPMs). We used machine learning classifiers available in Scikit-Learn: Classification Tree, k-NN, Logistic Regression, Random Forest, SVM, and MLP. The use of the CPM is necessary due to the low data regime (Xia et al., 2023). We focus on two CPMs: (i) Mol2Vec (Jaeger et al., 2018)[1] and (ii) the pre-trained JOINT TRANSFORMER (allowing us to evaluate our model in the role of a feature extractor). We run all baseline models with model selection to find the best hyperparameters.

**Results**   JOINT TRANSFORMER outperforms all baselines on all metrics (Table 2). In particular, JOINT TRANSFORMER correctly finds 73% of experimentally determined potent inhibitors for human APN, indicating excellent predictive performance in low data regimes. These results show that the JOINT TRANSFORMER framework improves upon CPM combined with standard classifiers and that it is a competitive model in a purely predictive task. Interestingly, there is no visible difference between using the two considered CPMs, i.e., Mol2Vec and a pre-trained JOINT TRANSFORMER. The pre-trained JOINT TRANSFORMER works better than Mol2Vec on, e.g., Logistic Regression, while for other baselines Mol2Vec results in better scores (e.g., k-NN, Random Forest).

Table 2: A comparison of various combinations of predictors and CPMs vs. JOINT TRANSFORMER.

| PREDICTOR | CPM | PRECISION | RECALL | F1 | ACCURACY |
|---|---|---|---|---|---|
| | MOL2VEC | 0.60 | 0.61 | 0.58 | 0.58 |
| CLASSIFICATION TREE | JOINT TRANSFORMER | 0.71 | 0.61 | 0.48 | 0.50 |
| | MOL2VEC | 0.72 | 0.73 | 0.69 | 0.69 |
| k-NN | JOINT TRANSFORMER | 0.66 | 0.65 | 0.58 | 0.58 |
| | MOL2VEC | 0.42 | 0.49 | 0.28 | 0.35 |
| LOGISTIC REGRESSION | JOINT TRANSFORMER | 0.64 | 0.58 | 0.46 | 0.48 |
| | MOL2VEC | 0.64 | 0.65 | 0.64 | 0.67 |
| RANDOM FOREST | JOINT TRANSFORMER | 0.71 | 0.63 | 0.50 | 0.52 |
| | MOL2VEC | 0.70 | 0.60 | 0.45 | 0.48 |
| SVM | JOINT TRANSFORMER | 0.56 | 0.56 | 0.52 | 0.52 |
| | MOL2VEC | 0.53 | 0.53 | 0.45 | 0.46 |
| MLP | JOINT TRANSFORMER | 0.57 | 0.56 | 0.47 | 0.48 |
| JOINT TRANSFORMER (OURS) | N/A | **0.73** | **0.75** | **0.72** | **0.73** |

## 4.3   DE NOVO DRUG DESIGN

**Task**   In the *de novo* drug design task, the goal is to generate valid molecules that maximize an objective function measuring how well a given molecule fulfills a desired property profile. For our task, we choose the multi-property objective functions from the GuacaMol benchmark (Brown et al., 2019): Perindopril MPO, Sitagliptin MPO, and Zaleplon MPO, which are also the three hardest to optimize (Gao et al., 2022). In practical applications, the evaluation budget (i.e., the maximum number of times the objective function can be evaluated) is a major bottleneck. We evaluate all methods on an evaluation budget equal to $10^3$, which is more practical than $10^4$ adopted by Gao et al. (2022). We note that in many applications a realistic number of evaluations is closer to $10^2$.

**Methods**   We compare the BBO with JOINT TRANSFORMER (Alg. 3.2) to other SMILES-based molecule optimization methods. In particular, we choose the three best-performing methods across all tasks in a benchmark comparing 25 various optimization methods (Gao et al., 2022): SMILES GA (Yoshikawa et al., 2018), REINVENT (Olivecrona et al., 2017) and an LSTM combined with a hill-climbing algorithm (LSTM + HC) (Brown et al., 2019). Additionally, we report the Dataset Best value, which is the best value of the objective function present in the dataset, as the upper bound for all screening methods, and MolPal (Graff et al., 2021), which is a deep-learning-based screening method. We include in the comparison the Variational Autoencoder (VAE) (Kingma & Welling, 2013; Rezende et al., 2014) combined with Bayesian optimization (VAE + BO) and a Junction Tree VAE

---

[1]Mol2Vec is a convolutional neural network for turning SMILES into vectors, available in DeepChem.

(Jin et al., 2018) combined with Bayesian optimization (JT-VAE + BO). Finally, we compare our method to a standard, unconditional decoder-only Transformer model, fine-tuned on examples with corresponding objective values above a fixed threshold $y_c \in \mathbb{R}$ (MolGPT + fine-tune).

**Results**   JOINT TRANSFORMER is the only method in this experiment that generates molecules better than in the dataset across all tasks, successfully performing *de novo* design (Table 3). The fine-tuned MolGPT is next in line in terms of performance. JOINT TRANSFORMER generates optimized molecules for an evaluation budget as low as 137, 452, and 17 evaluations, for the three optimization tasks respectively - outperforming other non-Transformer-based methods by a large margin. An additional investigation of the distribution of the objective function values for molecules sampled from the Transformer-based models as compared to the data distribution (Figure 1), shows that JOINT TRANSFORMER significantly alters the distribution of the sampled molecules towards optimal objective values, as opposed to MolGPT that only slightly skews the initial data distribution.

Table 3: The highest value of the objective function across all generated molecules (Top1) within $10^3$ evaluations. Mean and standard deviation across three independent data splits.

| METHOD | PERINDOPRIL MPO | SITAGLIPTIN MPO | ZALEPLON MPO |
|---|---|---|---|
| DATASET BEST | $0.53 \pm 0.00$ | $0.40 \pm 0.02$ | $0.50 \pm 0.01$ |
| MOLPAL | $0.49 \pm 0.01$ | $0.05 \pm 0.01$ | $0.16 \pm 0.09$ |
| SMILES GA | $0.44 \pm 0.01$ | $0.21 \pm 0.10$ | $0.28 \pm 0.08$ |
| REINVENT SMILES | $0.45 \pm 0.01$ | $0.02 \pm 0.01$ | $0.27 \pm 0.03$ |
| LSTM + HC | $0.47 \pm 0.01$ | $0.02 \pm 0.02$ | $0.14 \pm 0.03$ |
| VAE + BO | $0.45 \pm 0.01$ | $0.03 \pm 0.01$ | $0.04 \pm 0.04$ |
| JT-VAE + BO | $0.43 \pm 0.01$ | $0.05 \pm 0.03$ | $0.13 \pm 0.11$ |
| MOLGPT + FINE-TUNE | $0.54 \pm 0.03$ | $0.40 \pm 0.03$ | $0.51 \pm 0.00$ |
| JOINT TRANSFORMER (OURS) | $\mathbf{0.55 \pm 0.01}$ | $\mathbf{0.43 \pm 0.01}$ | $\mathbf{0.55 \pm 0.02}$ |

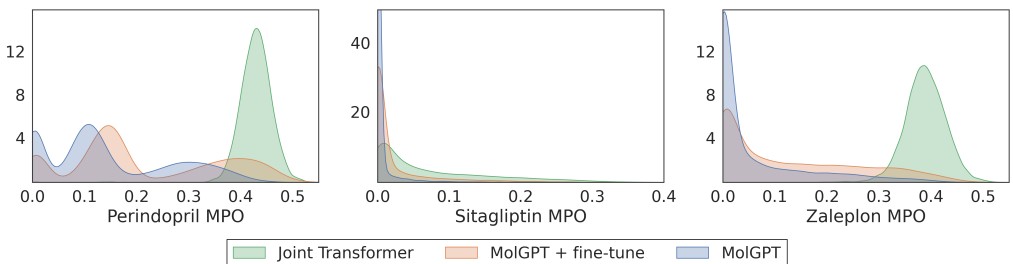

Figure 1: Distribution of the objective function values sampled from different models: JOINT TRANSFORMER (green), MolGPT + fine-tune (orange), MolGPT (blue) . Best viewed in color.

## 5   CONCLUSION

In this paper, we formulated the problem of de novo drug design as an instance of probabilistic BBO (Section 2.1). We proposed a general-purpose sampling algorithm that performs probabilistic BBO with any joint generative model (Section 3.2), with theoretical guarantees on the expected runtime as the function of the training data (Section 3.1.3). Finally, we proposed a joint generative model, called JOINT TRANSFORMER, that combines a Transformer decoder, a Transformer encoder, and a predictor in a single model with shared parameters (Section 3.1), which is jointly trained with a penalized log-likelihood objective (Eq 6). We empirically showed that JOINT TRANSFORMER successfully performs unconditional generation (Section 3.1.2), as it simultaneously samples valid new examples with corresponding target values (Section 4.1). We showcased the exceptional predictive performance of JOINT TRANSFORMER by its ability to identify potent inhibitors for human APN (Section 4.2). Finally, we showed that JOINT TRANSFORMER outperforms state-of-the-art approaches to *de novo* drug design (Section 4.3).

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
