## A  AUTHOR CONTRIBUTION

*Anonymized for the submission*

## B  PROOFS

### B.1  PROOF OF PROPOSITION 1

**Proposition 1.** *Let $p(\mathbf{x}, y)$ be a joint probability distribution over $\mathcal{X} \times \mathcal{Y}$. Let $y_c \in \mathcal{Y}$ be such that $p(y_c) > 0$. Then*

$$p(\mathbf{x} \mid y_c) \propto \mathbb{1}_{\{y=y_c\}}(y)p(y \mid \mathbf{x})p(\mathbf{x}).$$

*Proof.* Assume that $p(\mathbf{x}, y)$ is a joint probability distribution over $\mathcal{X} \times \mathcal{Y}$. Choose $y_{\max} \in Y$ to be such that $p(y \geq y_c) > 0$. Then a simple application of Bayes rule yields

$$p(\mathbf{x} \mid \{y \geq y_c\}) = \frac{p(\mathbf{x}, \{y \geq y_c\})}{p(\{y \geq y_c\})} = \frac{\mathbb{1}_{\{y \geq y_c\}}(y)p(y \mid \mathbf{x})p(\mathbf{x})}{p(\{y \geq y_c\})}. \tag{7}$$

Since $p(\{y \geq y_c\}) > 0$ and it does not depend on $\mathbf{x}$, we have that

$$p(\mathbf{x} \mid \{y \geq y_c\}) \propto \mathbb{1}_{\{y \geq y_c\}}(y)p(y \mid \mathbf{x})p(\mathbf{x}).$$

$\square$

### B.2  PROOF OF PROPOSITION 2

**Proposition 2.** *Let $p(y)$ be a probability distribution over $\mathcal{Y}$ with a corresponding cumulative distribution function $F$. Let target $y_c \in \mathcal{Y}$ be such that $p(y_c) > 0$ and let $p$ be the probability of sampling a target $y \sim p(y)$ such that $y > y_c$. The expected number of trials $N$ until obtaining a sample $y \sim p(y)$ such that $y > y_c$ is equal to $1/p$.*

*Proof.* Let $p(y)$ be a probability distribution over $\mathcal{Y}$ with a corresponding cumulative distribution function $F$. Let $y_c \in \mathcal{Y}$ be such that $p(y_c) > 0$. Define r.v. $N$ as the number of trials until obtaining a sample $y > y_c$, where $y$ is distributed as $p(y)$. For each $n \in \mathbb{N}$, the distribution of $N$ is given by

$$P(N = n) = (1 - p)^{n-1}p,$$

where $p = 1 - F(y \leq y_c)$. Hence, the number of trials $N$ follows a geometric distribution with an expected value equal to $\mathbb{E}[N] = 1/p$. $\square$

## C  ADDITIONAL EXPERIMENTS

### C.1  MOLECULE GENERATION

**Task**   In the molecule generation task, the goal is to generate valid and novel molecules that follow the chemical distribution of the training data. Following Brown et al. (2019), we evaluate all molecule generation methods on five metrics: validity, a fraction of the generated molecules that are correspond to a valid SMILES string; uniqueness, a fraction of the generated molecules that are unique; novelty, a fraction of the generated molecules that are not present in the training data; KL Divergence, a measure of similarity of the generated molecules to the training set with respect to selected chemical properties (Brown et al., 2019), as well as Fréchet ChemNet Distance (FCD; (Preuer et al., 2018)), a general measure of similarity of the generated molecules to the training set.

**Baselines**   As baselines, we select well-established molecule generation models based on SMILES representation (Weininger, 1988): LSTM (Ertl et al., 2018), VAE (Kingma & Welling, 2013; Rezende et al., 2014) and AAE (Kadurin et al., 2016). Additionally, we consider graph-based models: Junction Tree VAE (Jin et al., 2018), MoLeR (Maziarz et al., 2021) and MAGNet (Hetzel et al., 2023). Finally, we include MolGPT (Bagal et al., 2022), which is a Transformer-based model and the backbone for the JOINT TRANSFORMER, sharing the same architecture, but trained differently.

**Results** In the molecule generation task, JOINT TRANSFORMER successfully generates valid, unique and novel molecules (Tab. 4). Moreover, JOINT TRANSFORMER generates molecules with properties that closely follow the training set distribution, making the newly generated molecules realistic and physio-chemically plausible, as measured by KL Divergence and FCD. Compared to the backbone MolGPT model, JOINT TRANSFORMER achieves identical performance, showing that the modified training procedure does not hurt the generative functionality of the model. From the generative modeling perspective, this result is counterintuitive, as we can include the reconstruction task to the training procedure of the JOINT TRANSFORMER, without sacrificing its generative performance.

Overall, none of the molecule generation methods achieves best performance across all metrics. Graph-based methods outperform others on validity, as they generate always valid molecules by design. However, the improvement of 3% as compared to Transformer-based models (JOINT TRANS-FORMER and MolGPT) is negligible. Additionally, it comes at the expense of generating molecules with decreased (from 12% to 19%) values of the KL Divergence and FCD metrics. On the other hand, LSTM achieves top performance on KL Divergence and FCD metrics, slightly (1% and 3%, respectively) outperforming Transformer-based methods, but falls behind in the validity of the generated molecules. All methods successfully generate unique and novel molecules. Overall, JOINT TRANSFORMER strikes a good balance between graph-based and SMILES-based LSTM, making it a viable choice for a go-to molecule generation model.

Table 4: Molecule Generation Task. JOINT TRANSFORMER (JT) matches state-of-the-art performance of different molecule generation methods. Training the JOINT TRANSFORMER model on generation and reconstruction tasks simultaneously does not hurt the generation performance of the model.

| MODEL | SIZE | VALIDITY ($\uparrow$) | UNIQUENESS ($\uparrow$) | NOVELTY ($\uparrow$) | FCD ($\uparrow$) | KL DIV ($\uparrow$) |
|---|---|---|---|---|---|---|
| LSTM | - | 0.96 | **1.0** | 0.91 | **0.91** | **0.99** |
| VAE | - | 0.87 | **1.0** | 0.97 | 0.86 | 0.98 |
| AAE | - | 0.82 | **1.0** | **1.0** | 0.53 | 0.89 |
| JT-VAE | - | **1.0** | N/A | N/A | 0.76 | 0.94 |
| MAGNET | 6.9M | N/A | N/A | N/A | 0.73 | 0.92 |
| MOLER | - | **1.0** | 0.99 | 0.97 | 0.78 | 0.98 |
| MOLGPT | 6M | 0.98 | **1.0** | **1.0** | **0.91** | **0.99** |
| MOLGPT (OURS) | 6M | 0.97 | **1.0** | 0.97 | 0.89 | 0.98 |
| JT (OURS) | 6M | 0.97 | **1.0** | 0.98 | 0.89 | **0.99** |
| JT (OURS) | 50M | 0.98 | **1.0** | 0.95 | 0.90 | **0.99** |

## C.2 UNCONDITIONAL GENERATION

Moreover, the jointly trained predictor $q_{\theta,\phi}(y \mid \mathbf{x})$ of the JOINT TRANSFORMER generalizes well to data generated with the model $p_\theta(\mathbf{x})$. In particular, the prediction error, as measured by mean absolute error, of the JOINT TRANSFORMER fine-tuned on three properties from the Guacamol task (Brown et al., 2019) do not change between the test set and newly generated data (Table 5). This shows good generalization performance of JOINT TRANSFORMER.

Table 5: Mean absolute prediction error (MAE) for the predictor on three property prediction tasks on test and generated data. Mean and standard deviation across independent runs.

| METHOD | DATA | PERINDOPRIL MPO | SITAGLIPTIN MPO | ZALEPLON MPO |
|---|---|---|---|---|
| JOINT TRANSFORMER | TEST | $0.014 \pm 0.004$ | $0.009 \pm 0.001$ | $0.012 \pm 0.001$ |
|  | GENERATED | $0.015 \pm 0.004$ | $0.009 \pm 0.002$ | $0.012 \pm 0.001$ |

# D  IMPLEMENTATION DETAILS

## D.1  DATA AND TOKENIZATION

We use SMILES (Weininger, 1988) based representations of molecules across all experiments. In all experiments we pre-train the JOINT TRANSFORMER in an unsupervised manner using the ChEMBL database, a manually curated database of molecules with drug-like properties (Mendez et al., 2019). As opposed to other datasets like ZINC (Irwin et al., 2020), ChEMBL contains only molecules which have been synthesized. To ensure reproducibility and comparability with molecule generation baselines we use version 24 of the database that contains 1.8M compounds altogether and apply standard data processing used in the Guacamol benchmark (Brown et al., 2019). For supervised finetuning in Sections 4.1 and 4.2, we randomly select a subset ($N = 1000$) of the unsupervised data and evaluate the objective functions on the selected subsets. As for tokenization of the data, we use a tokenizer based on (Schwaller et al., 2020). We additionally use an augmentation method of SMILES representations based on (Tetko et al., 2019) and similar to (Bagal et al., 2022) across all experiments and methods. This ensures transferability of results obtained by Bagal et al. (2022) to our experiments.

## D.2  MOL2VEC PRE-TRAINING DATA

The corpus of compounds was composed using the ZINC v15 and ChEMBL v23 databases as source of compounds. The two databases were merged, duplicates removed, only compounds kept that could be processed by RDKit, and filtered using the following cutoffs and criteria: molecular weight between 12 and 600, heavy atom count between 3 and 50, clogP between -5 and 7, and only H, B, C, N, O, F, P, S, Cl, Br atoms allowed. Additionally, all counter ions and solvents were removed and canonical SMILES generated by RDKit. This procedure yielded 19.9 million compounds.

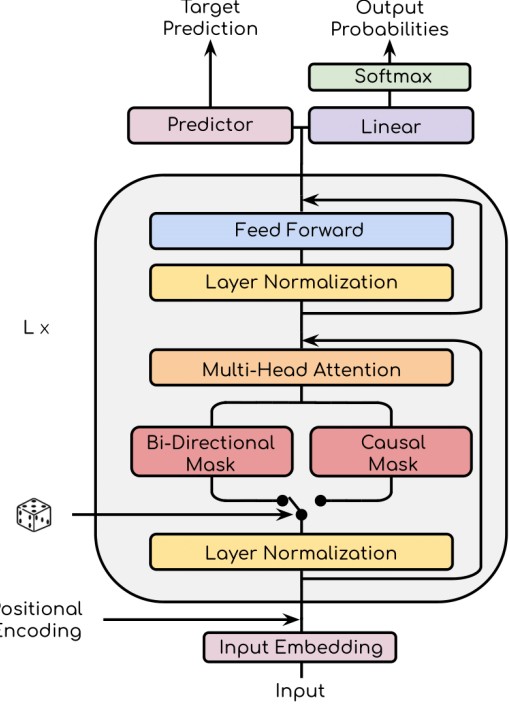

Figure 2: JOINT TRANSFORMER architecture.

### D.3 ARCHITECTURE

Our implementation of the JOINT TRANSFORMER follows the implementation provided by (Karpathy, 2023), which is a re-implementation of a GPT-2 (Radford et al., 2019) used by MolGPT (Bagal et al., 2022). The only difference is that during each forward pass, we switch between a causal and a bidirectional masking, depending on the task we are optimizing for. We additionally stack an MLP network on the top of the first output token for prediction. The complete list of hyperparameters is presented in Table 6. Our implementation results in a model with 6.5M parameters.

Table 6: Model hyperparameters for the JOINT TRANSFORMER used across all experiments.

| HYPERPARAMETER | VALUE |
|---|---|
| ACTIVATION FN | GELU |
| EMBED DIM | 256 |
| NUM LAYERS | 6 |
| NUM HEADS | 8 |
| FEEDFORWARD DIMENSION | 1024 |
| FEEDFORWARD BIAS | FALSE |
| LAYER NORM EPSILON | $1e{-}5$ |
| PREDICTOR HEAD | MLP |
| PREDICTOR NUM LAYERS | 1 |
| PREDICTOR HIDDEN DIM | 100 |

### D.4 TRAINING

The JOINT TRANSFORMER can be trained in an unsupervised, semi-supervised or supervised setting, depending whether a target $y \in \mathcal{Y}$ is sampled from the dataset $\mathcal{D}$ or is not available. We show the unsupervised training procedure for JOINT TRANSFORMER in Algorithm 3.

---

**Algorithm 3** Unsupervised training of JOINT TRANSFORMER

---

**Input:** A dataset $\mathcal{D} = \{\mathbf{x}_n\}_{n=1}^{N}$. JOINT TRANSFORMER $p_{\theta,\phi}(\mathbf{x}, y)$ with parameters $\theta, \phi$ containing a decoder $p_\theta(\mathbf{x})$, encoder $\prod_{d=1}^{D} p_\theta(x_d \mid \mathbf{m} \odot \mathbf{x}_{-d})$ and a predictor $p_{\theta,\phi}(y \mid \mathbf{x})$.
Task probability $p_{\text{task}} \in [0, 1]$ and a masking distribution $q(\mathbf{m})$.
1: **while** a stopping criterion is not met **do**
2:     Uniformly sample $\mathbf{x}$ from the dataset $\mathcal{D}$
3:     Sample an indicator $u \sim \text{BERNOULLI}(p_{task})$
4:     **if** $u = 0$ **then**
5:         Sample mask $\mathbf{m} \sim q(\mathbf{m})$
6:         Calculate loss $\ell(\theta, \phi) = -\sum_{d=1}^{D} \ln p_\theta(x_d \mid \mathbf{m} \odot \mathbf{x}_{-d})$
7:     **else**
8:         Set mask to the causal mask
9:         Calculate loss $\ell(\theta, \phi) = -\ln p_\theta(\mathbf{x})$
10:     **end if**
11:     Update parameters $\theta, \phi$ using an optimizer w.r.t. loss $\ell$
12: **end while**

---

We provide the complete list of hyperparameters used for training JOINT TRANSFORMER in Table 7. JOINT TRANSFORMER was trained on a single NVIDIA GeForce RTX 2080 TI GPU for 4.2M iterations that took approximately seven days.

### D.5 FINE-TUNING

As JOINT TRANSFORMER is a joint model, fine-tuning is achieved by standard training (Alg. 1) on the supervised part of the dataset. Unless stated otherwise, we use the same set of hyperparameters for fine-tuning across all tasks, summarized in Table 8. Fine-tuning on a single NVIDIA GeForce

Table 7: Training hyperparameters of the JOINT TRANSFORMER used across all experiments.

| HYPERPARAMETER | VALUE |
|---|:---:|
| BATCH SIZE | 64 |
| TOTAL NUMBER OF TRAINING ITERATIONS | 4.2 M |
| OPTIMIZER | ADAMW |
| WEIGHT DECAY | 1e−1 |
| BETA 1 | 0.9 |
| BETA 2 | 0.95 |
| MAXIMUM LEARNING RATE | 6e−4 |
| MINIMUM LEARNING RATE | 6e−5 |
| DECAY LEARNING RATE | TRUE |
| WARMUP ITERATIONS | 2000 |
| NUMBER OF LEARNING RATE DECAY ITERATIONS | 4.2 M |
| VALUE TO CLIP GRADIENTS AT | 1.0 |
| DROPOUT | 0.1 |
| TASK PROBABILITY $p_{\text{task}}$ | 0.95 |

RTX 2080 TI GPU for 50K iterations takes approximately an hour. Hyperparameters not listed in Table 8 are shared with the pre-training task.

Table 8: Fine-tuning hyperparameters for the JOINT TRANSFORMER used across all experiments.

| HYPERPARAMETER | VALUE |
|---|:---:|
| DECAY LR | FALSE |
| LEARNING RATE | 3e−5 |
| NUM OF ITERATION | 50K |
| TASK PROBABILITY $p_{\text{task}}$ | 0.1 |