# OpenReview forum: "De Novo Drug Design with Joint Transformers"
_ICLR.cc/2024/Conference — ICLR 2024 Conference Withdrawn Submission_

### Official Review · Reviewer_jRVR · 2023-10-26

**Soundness:** 2 fair
**Presentation:** 2 fair
**Contribution:** 1 poor
**Rating:** 3
**Confidence:** 4

**Summary:**

The paper addresses the de novo molecule design problem by employing a joint transformer that consists of an encoder, a decoder, and a predictive network. Moreover, they also propose a black-box optimization algorithm that is able to generate new molecules with improved proproteis.

**Strengths:**

1. The paper tackles an important problem and the chosen properties to condition on are also interesting.

**Weaknesses:**

1. The novelty is not very strong
2. There are many baselines are missing

**Questions:**

1. The paper a bit undermines the existing works that are already developed to tackle the conditional generation directly without relying on extra black box optimizers such as Bayesian optimization.  Works such as
   1)Conditional Molecular Design with Deep Generative Models, https://arxiv.org/pdf/1805.00108.pdf
   2) MolGPT: Molecular Generation Using a Transformer-Decoder Model, which is mentioned in the paper and pointed out that the model
      lacks a predictor, but if the goal is the conditional generation, and if we have enough labeled data (which was also used in this paper to
      learn the predictor), where we can train a conditional generative model that later can be used to generate molecules with the target
      property, do we still need a predictor during the generation which we can always obtain by separate training if we want to evaluate the
      newly generated molecules property.
 3) Conditional generation of molecules from
     disentangled representations https://ml4molecules.github.io/papers2020/ML4Molecules_2020_paper_52.pdf
     and many others in recent years, there are many papers that directly tackle conditional generation without the need for extra
     optimization.
There are many baselines shown in Guacamol paper, for instance, for the goal directed generating, for the ZALEPLON MPO score, the highest performance reached 0.754 from graphGA, but this baseline did not appear in the paper.


2. The baselines for the property predictor are very weak, in recent years there have been tons of papers published for molecule property prediction. If we skip all the papers that actually use graphs or 3D structures, the models that are trained on SMILES representation which have a predictor are also missing for instance:
Automatic Chemical Design Using a Data-Driven Continuous Representation of Molecules

It is true that there are too many baselines exist around the topic, it is impossible to compare all of them but a good selection of recent work would needed to make the point that the current model is superior to the existing ones.

3. I am a bit confused about model architecture, was wondering how the parameter sharing happens between encoder and decoder, the parameters are fully shared or only part of them are shared? A figure explaining the model architecture would have been helpful.

4. The probabilistic black-box optimization method proposed in algorithm 2 seems to me more like a  filtering process but not really an optimization, unless I have misunderstood.
5. The proofs for the proposition are said provided in the appendix but there was no appendix is available.

---

### Official Review · Reviewer_FkeG · 2023-10-30

**Soundness:** 2 fair
**Presentation:** 2 fair
**Contribution:** 1 poor
**Rating:** 3
**Confidence:** 4

**Summary:**

This paper proposes a transformer-based, encoder-decoder framework for conditional molecule generation and property prediction. and several experiments demonstrate that this framework achieves SOTA in targeted virtual screening and de novo drug design.

**Strengths:**

Formulate a generic sampling algorithm with theoretical guarantees to guide the generation of novel compounds with this methods.

**Weaknesses:**

1. The experiments and benchmarks presented are not very convincing. In targeted virtual screening, this is a typical problem. It would be better to use DUDE or PCBA as benchmark datasets, also the baseline methods are not highly comparable, lacking pretraining or deep learning-based methods. In de novo drug design, it is also necessary to compare with recent deep learning-based state-of-the-art methods, including popular VAE, GAN, diffusion, and flow matching-based models, such as G-SchNet and EDM.

2. I disagree with the approach of treating molecule generation as BBO problems, as molecules should be calculated accurately using first-principle methods, such as DFT and MD. The author should clarify their reasoning for considering this as a BBO problem.

3. The statement, `it is probabilistic and therefore tailored to avoid non-realistic molecules as sampled examples.` is unclear as to why this sampling strategy can lead to generating realistic molecules. Further discussion and experiments are needed to emphasize this point.

**Questions:**

I believe that incorporating a framework or workflow diagram into the author's writing would greatly enhance the clarity and organization of the text.

---

### Official Review · Reviewer_LFPx · 2023-10-31

**Soundness:** 2 fair
**Presentation:** 3 good
**Contribution:** 3 good
**Rating:** 5
**Confidence:** 4

**Summary:**

The paper proposed a method that uses generative transformer architecture for $\textit{de novo}$ drug design. To achieve the goal, the authors use a joint transformer architecture which combines a transformer encoder, a transformer decoder, and a predictive head. The encoder and decoder share the same weights and the only difference is the mask applied to the weights during training and inference. The joint transformer model is trained with penalized negative log-likelihood loss function with a probability parameter to shift between mask language modeling + regression loss and autoregressive loss. The authors claim that the joint transformer model can reach high accuracy in prediction task for virtual screening, and outperforming SOTA optimization methods in $\textit{de novo}$ drug design tasks.

**Strengths:**

1. The share weight design of the encoder and decoder enables the model to learn robust representation of molecules for both target prediction and SMILES sequence generation. It can also lead to more computationally efficient model, as claimed by the authors.

2. The design of training the joint transformer using a probability hyperparameter to shift between encoder and decoder mode is interesting. It can also cast influence on future works in similar tasks.

3. The manuscript is written with high clarity and easy to follow proofs.

**Weaknesses:**

1. The joint transformer design adds complexity to the training. Despite the advantages mentioned in the Strength section, the three terms (penalty, prediction loss, generative loss) in the loss function needs extra heuristic hyperparameter tuning ($p_{task}$). From the result of Table 1, the choice of $p_{task}$ and the penalty term in loss function will result in trade-off between Validity/FCD and the prediction accuracy of the model.

2. According to Algorithm 2, the probabilistic black-box optimization follows a rejection sampling type of method: if the predicted value of a generate molecule is not higher than a pre-defined threshold, the generated molecule is rejected. I am wondering if such a sampling method will affect the efficiency of the algorithm, especially when the prediction part of the model is undertrained (e.g. when labeled data of a desired property is limited).

**Questions:**

1. It seems like the property prediction head can only be used to sample a label $y$ from $x$ due to the joint training loss. Is there any deterministic way of predicting the label? What is the loss to train the predictive MLP head? If the MLP head predicts a single value, how is it a sampling process?

2. Page 6, “it is probabilistic and therefore tailored to avoid non-realistic molecules as sampled examples”. This claim seems odd to me as a probabilistic model can sample non-realistic molecules. This is also reflected from the results of the manuscript, like Table 1, in which the validity of generated molecules is not 1.

3. Section 4.2, what’s the purpose of virtual screening benchmarking? Also, the baseline is weak from my perspective because in low data regime, simple ECFP with XGBoost/LightGBM can be the best model compared to pretrained learned features.

4. Some important previous works for $\textit{de novo}$ drug design are missing such as arxiv.org/abs/2206.09010 and www.science.org/doi/10.1126/sciadv.aap7885

---

### Official Review · Reviewer_etCU · 2023-10-31

**Soundness:** 1 poor
**Presentation:** 2 fair
**Contribution:** 2 fair
**Rating:** 3
**Confidence:** 4

**Summary:**

The core contribution of this paper is proposing JOINT TRANSFORMER, combining a Transformer decoder, a Transformer encoder, and a predictor with shared parameters. With theoretical basis, *de novo* drug design is formulated as a probabilistic black-box optimization problem, and a general-purpose sampling algorithm is proposed. The authors use some experiments to show the performance of their approach.

**Strengths:**

## Strengths

1. Formulating *de novo* drug design as a probablistic BBO problem is a novel perspective.
2. The writing of the paper is smooth, logical and clear, especially the theoretical formulation is solid.

**Weaknesses:**

## Weaknesses

### Related works

Chemformer (https://iopscience.iop.org/article/10.1088/2632-2153/ac3ffb/pdf) also incorporates a bidirectional encoder and an autoregressive decoder to process SMILES. It seems that the model architectures of JOINT TRANSFORMER and Chemformer are similar, but the authors make no mention of this.

### Experiments on targeted virtual screening (section 4.2)

**This part of experiments cannot prove the effectiveness of the algorithm in virtual screening tasks.** The author seems completely unaware of the important baselines and evaluation ways of virtual screening. Please refer to https://pubs.acs.org/doi/10.1021/acs.jcim.5b00090, https://arxiv.org/pdf/2310.06367.pdf.

In addition, the model for virtual screening can be separated from the model for drug design (molecular generation). This paper should focus on experiments on drug design tasks.

### **Experiments on *de novo* drug design (section 4.3)**

As indicated by the title, experiments on *de novo* drug design is the most important part to demonstrate the effectiveness of JOINT TRANSFORMER. However, experiments are only conducted on three tasks selected from the GuacaMol dataset, and there are obvious problems with the experimental setup.

1. The authors claim that they choose three tasks "that are hardest to optimize" from the GuacaMol benchmark, but there is no basis for this. The GuacaMol benchmark for de novo drug design contains 20 goal-directed tasks, and Gao et al. (2022) also conduct experiments on all the 20 tasks. To the best of my knowledge, no previous work has come to the conclusion that these three MPO tasks are the hardest in GuacaMol. (Note: Low scores do not necessarily mean difficulty, as the score distribution is different for different tasks.)

   **The experiments should be conducted on all 20 tasks in the GuacaMol benchmark.**

2. The evaluation budget is set to 1000, and the authors claim that "this is more practical than 10000", "in many applications a realistic number of evaluations is closer to 100", but no evidence is provided. Actually, the official setting of the GuacaMol benchmark does not limit the number of evaluations at all, and in most recent works on computer-aided drug design, the number of evaluations is larger than 10000, thanks to the development of computing tools such as docking software.

   **It doesn't make sense to limit the number of evaluations to 1000, and the authors should follow the official settings of GuacaMol.** (Gao et al. (2022) focuses on sample efficiency, but this is not mainstream practice.)

3. The authors only report the highest score (top-1) of the generated molecules, which is different from the official setting of the GuacaMol benchmark (average of top-1, top-10 and top-100 scores).

   **Similarly, the authors should follow the official settings of GuacaMol.**

4. The authors claim that "JOINT TRANSFORMER outperforms state-of-the-art approaches" without comparisons to some important recent baselines, such as https://arxiv.org/pdf/2007.04897.pdf.

5. One minor problem: The results of GuacaMol experiments are usually expressed in three decimals, but two decimals are used in this paper.

6. "Designing molecules that dock well" has been recognized as the key criterion for algorithms for drug design (https://pubs.acs.org/doi/epdf/10.1021/acs.jcim.2c01355), and most recent works conduct experiments on docking (https://arxiv.org/pdf/2206.09010.pdf). So, I suggest the author add docking experiments.

In short, **this part of the experiments cannot fully support the effectiveness of JOINT TRANSFORMER on *de novo* drug design**, which is the major concern in this paper.

**Questions:**

## Questions

1. What if I use the scoring function to replace the predictor, as they are more accurate? In this paper there is no experiment on property prediction, so I doubt the effectiveness of the predictor.
2. I am puzzled by the PBBO algorithm in section 3.2. In real-world *de novo* drug design tasks, there may not be any molecules in the prior distribution that satisfy the property conditions, i.e. $p=0$ in Proposition 2 (that is what *de novo* means). In this case, JOINT TRANSFORMER will not work at all.
3. I am not sure if this is a typo, but there is no update to the model parameters in Proposition 2, even if $y_i\geq y_c$.